# Contextual Object Grouping (COG): A Specialized Framework for Dynamic Symbol Interpretation in Technical Security Diagrams

Jan Kapusta *, Waldemar Bauer and Jerzy Baranowski *

Department of Automatic Control and Robotics, AGH University of Krakow, 30-059 Krakow, Poland; bauer@agh.edu.pl

* Correspondence: jkapusta@agh.edu.pl (J.K.); jb@agh.edu.pl (J.B.)

**Abstract**

This paper introduces Contextual Object Grouping (COG), a specific computer vision framework that enables automatic interpretation of technical security diagrams through dynamic legend learning for intelligent sensing applications. Unlike traditional object detection approaches that rely on post-processing heuristics to establish relationships between the detected elements, COG embeds contextual understanding directly into the detection process by treating spatially and functionally related objects as unified semantic entities. We demonstrate this approach in the context of Cyber-Physical Security Systems (CPPS) assessment, where the same symbol may represent different security devices across different designers and projects. Our proof-of-concept implementation using YOLOv8 achieves robust detection of legend components (mAP50 $\approx$ 0.99, mAP50–95 $\approx$ 0.81) and successfully establishes symbol–label relationships for automated security asset identification. The framework introduces a new ontological class—the contextual COG class that bridges atomic object detection and semantic interpretation, enabling intelligent sensing systems to perceive context rather than infer it through post-processing reasoning. This proof-of-concept appears to validate the COG hypothesis and suggests new research directions for structured visual understanding in smart sensing environments, with applications potentially extending to building automation and cyber-physical security assessment.

**Keywords:** Contextual Object Grouping; security diagrams; symbol interpretation; object detection; semantic grouping; Cyber-Physical Security Systems; dynamic legend learning; intelligent sensing

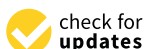

## 1. Introduction

*1.1. Problem Motivation: The Challenge of Symbol Standardization in Security Diagrams*

In the assessment of Cyber-Physical Security Systems (CPPS), one critical component involves the automatic detection and identification of security elements (cameras, sensors, access control devices) within building floor plans and security diagrams [1]. This process requires not only detecting the presence of security symbols but also correctly interpreting their meaning and establishing their relationship with technical specifications stored in separate databases. The ultimate goal is to create a comprehensive security asset inventory that maps each detected device to its physical location, technical parameters (detection range, security class, age), and potential impact on adversarial attack paths.

While the fusion of detected elements with external technical databases appears relatively straightforward, the automatic detection and interpretation of security elements



presents significant challenges rooted in the fundamental evolution of computer vision. The field has progressed from classical feature-based approaches like SIFT [2] through CNN breakthroughs including AlexNet [3], VGG [4], and ResNet [5], to modern object detection frameworks such as R-CNN [6], YOLO [7], and recent vision–language models [8]. However, despite these advances, the core difficulty lies in the lack of symbol standardization across different designers and projects. A red circle may represent a camera in one designer's legend, while another designer uses the same symbol to denote a PIR sensor or electromagnetic lock. This variability makes traditional object detection approaches, which rely on learning fixed symbol-to-meaning mappings, fundamentally inadequate for real-world deployment.

Recent advances in adaptive cyber-defense provide relevant context for COG's perceptual contributions. Tan et al. [9] survey how Moving Target Defense can be formulated in game-theoretic terms, highlighting strategy adaptation over time and state. Complementarily, evolutionary time-delay honeypot models [10] demonstrate how learning dynamics optimize responses to adversaries. These studies emphasize that modern defense requires accurate, context-driven asset semantics, precisely what COG provides through legend-as-grounding in technical diagrams.

### 1.2. Limitations of Traditional Approaches and the Nuance of Modern Contextual Models

Conventional object detection systems excel at identifying individual entities but struggle to capture semantics arising from structured arrangements of those entities [11,12]. Standard approaches typically follow a two-stage pipeline:

1. First detecting individual objects (symbols, labels, geometric elements);
2. Then, applying post-processing heuristics to infer relationships and establish meaning.

In the context of security diagrams, this approach faces several critical limitations:

- **Symbol Ambiguity:** The same visual symbol may represent completely different security devices across different design standards and individual preferences.
- **Brittle Post-Processing:** Rule-based heuristics for connecting symbols to their meanings are domain-specific, difficult to generalize, and prone to failure when encountering new design styles.
- **Context Separation:** Traditional pipelines separate perception (object detection) from interpretation (relationship inference), creating a semantic gap that is difficult to bridge reliably.
- **Scalability Issues:** Each new design style or symbol convention requires manual rule engineering, making the system difficult to scale across diverse security diagram formats.

Recent advances in contextual object detection, including YOLO-World [8] with its vision–language integration and Florence-2 [13] with its unified multi-task capabilities, demonstrate significant progress in contextual understanding. These models leverage large-scale pre-training on diverse image and text datasets to achieve impressive open-vocabulary object detection and multi-tasking performance. YOLO-World, for instance, allows for real-time open-vocabulary detection by integrating vision and language embeddings directly into the detection pipeline. Florence-2 pushes this further by unifying various vision tasks, including detection, segmentation, and captioning, under a single representation, enhancing its ability to generate rich, context-aware outputs.

However, despite these advancements, these approaches still operate primarily on pre-defined or large-scale, general-purpose object vocabularies and struggle with the dynamic semantic mapping required in technical diagram interpretation. While they excel at associating common objects with their textual descriptions (e.g., identifying a

"car" or "person"), they are not inherently designed to learn concrete, project-specific symbol–meaning relationships from a contained legend on the fly. Specifically:

- **Reliance on Pre-defined Vocabularies/Large-Scale Knowledge:** YOLO-World and Florence-2, while "open-vocabulary," infer meaning based on broad pre-trained knowledge. They are adept at recognizing objects that have been extensively represented in their training data or can be logically inferred from existing vocabulary. In contrast, technical diagrams often use highly abstract or non-standard symbols whose meaning is exclusively defined within the diagram's accompanying legend. A "red circle" can mean anything, and its meaning cannot be guessed from general world knowledge.
- **Post Hoc Interpretation of Specific Symbols:** For symbols unique to a specific diagram, these advanced models would still typically detect the visual primitive (e.g., "circle," "arrow") and then require a subsequent, separate process to link this primitive to its specific, dynamic meaning as defined in the legend. This reintroduces the semantic gap that COG aims to eliminate. The interpretation of the legend itself—identifying symbol–label pairs as functional units—appears not to be a first-class objective for these general-purpose models.
- **Lack of Explicit "Legend-as-Grounding" Mechanism:** While vision–language models can process text, they do not inherently possess a mechanism to treat a specific section of an image (the legend) as the definitive, dynamic ground truth for interpreting other visual elements within that same document. COG, by contrast, elevates the legend's symbol–label pairing into a first-class detectable contextual object (Row_Leg). This allows the model to learn the compositional grammar of the diagram directly from the legend, making the interpretation process intrinsically linked to the diagram's self-defining context.

In essence, while modern vision–language models are powerful in general contextual understanding, they are primarily focused on vocabulary expansion and robust detection of individual entities or generic relationships. They do not intrinsically perceive and learn structured, dynamic semantic groupings that emerge from the unique, localized "visual language" of a technical diagram's legend. COG specifically addresses this gap by enabling models to directly learn and detect these contextual groupings as unified semantic entities, fundamentally shifting from "detect then reason" to "perceive context directly" within the specialized domain of technical diagrams.

Unlike existing approaches that treat symbol–text relationships as post-processing tasks, COG appears to fundamentally shift the paradigm by making contextual groupings first-class detection targets. This seems to address a critical gap where advanced models like YOLO-World and Florence-2, despite their impressive contextual capabilities, cannot dynamically learn project-specific symbol semantics from contained legends within documents. While these models excel at leveraging pre-trained knowledge or large-scale vocabularies, they appear to lack the mechanism to treat diagram-specific legends as definitive, dynamic ground truth for interpreting visual elements within the same document. COG specifically addresses this limitation by elevating legend-based symbol–label pairings into directly detectable contextual objects, potentially enabling models to learn the compositional grammar of technical diagrams on-the-fly.

It is important to clarify that COG serves as a specialization framework that promotes contextual groupings (e.g., symbol–label pairs) to first-class detection targets, rather than modifying existing detectors like YOLO. In this proof-of-concept, YOLOv8m provides a convenient single-stage architecture to demonstrate joint learning of atomic and COG classes.

*1.3. The COG Solution: Context as Perception*

To address these limitations, we propose Contextual Object Grouping (COG), a framework that fundamentally shifts the approach from "detect then reason" to "perceive context directly." COG enables models to learn and detect contextual groupings such as symbol–label pairs in a legend, as unified semantic entities rather than separate objects requiring post hoc relationship inference.

The key insight underlying COG is that in technical diagrams, meaning often emerges not from individual symbols but from their structured co-occurrence with contextual elements, particularly legend entries that define symbol semantics. A red circle is semantically meaningless in isolation; it becomes a "camera" only when paired with its corresponding label in the legend. COG captures this relationship by training models to directly detect these Symbol + Label pairs as first-class contextual objects, building upon foundational work in semantic segmentation [12] and structured scene understanding [14].

This approach aligns with recent developments in document understanding, where LayoutLMv3 [15] and DocLLM [16] demonstrate the importance of spatial–semantic relationships. However, unlike these text-centric approaches, COG focuses on visual contextual groupings in technical diagrams, creating a new paradigm for intelligent sensing applications.

*1.4. Contributions*

This paper makes the following contributions:

- **Specific Ontological Framework:** We introduce the concept of contextual COG classes as a new intermediate ontological level between atomic object detection and high-level semantic interpretation, supported by comprehensive evaluation using established metrics [17].
- **Dynamic Symbol Interpretation:** We demonstrate how COG enables automatic adaptation to different symbol conventions through legend-based learning, eliminating the need for symbol standardization in security diagram analysis.
- **Proof-of-Concept Implementation:** We present a working system using YOLOv8 [18] that achieves high accuracy in detecting and structuring legend components in security diagrams, with performance validation following established benchmarking practices [17].
- **Hierarchical Semantic Structure:** We show how COG can construct multi-level semantic hierarchies (Legend → COG(Row_Leg) → Symbol + Label) that capture the compositional nature of technical diagrams.
- **Practical Application Framework:** We demonstrate the application of COG to real-world security assessment tasks, showing how detected elements can be integrated with technical databases for comprehensive asset analysis in intelligent sensing environments [19].

## 2. Related Work

*2.1. Object Detection Evolution: From Classical to Modern Approaches*

The evolution of object detection provides essential context for understanding COG's contributions. Classical approaches relied on hand-crafted features, with SIFT [2] establishing scale-invariant feature extraction principles that influenced later developments. The deep learning revolution began with AlexNet [3], which demonstrated the power of convolutional neural networks for image classification, followed by architectural improvements in VGG [4] and ResNet [5].

Modern object detection emerged with R-CNN [6], which introduced region-based detection through selective search and CNN feature extraction. This evolved through

Fast R-CNN [20] and Faster R-CNN [11], culminating in two-stage detection frameworks. YOLO [7] revolutionized the field by formulating detection as a single regression problem, directly predicting class probabilities and bounding box coordinates. DETR [21] represented a paradigm shift by leveraging transformers to perform detection as a direct set prediction problem, eliminating anchor boxes and non-maximum suppression.

Recent advances include YOLO-World [8], which achieves 35.4 AP at 52.0 FPS through vision–language integration, and DetCLIPv3 [22], demonstrating versatile generative open-vocabulary detection. These developments show progress toward contextual understanding, but they focus on vocabulary expansion rather than the structural semantic groupings that COG addresses.

Despite their architectural differences, all these detectors focus on identifying and classifying individual objects. While they may utilize contextual visual features implicitly through convolutional receptive fields or transformer attention mechanisms, they do not explicitly encode or learn object co-occurrence as structured entities. COG extends these approaches by formalizing co-occurrence patterns as learnable object classes.

### 2.2. Semantic Segmentation and Structured Understanding

Semantic segmentation, pioneered by Fully Convolutional Networks (FCN) [12], established pixel-level understanding that informs spatial relationships. U-Net [23] advanced this through encoder–decoder architectures, while Mask R-CNN [24] demonstrated simultaneous detection and segmentation capabilities that relate to COG's contextual understanding approach.

Recent work in panoptic segmentation [14] and scene graph generation provides relevant context. 4D Panoptic Scene Graph Generation [25] introduces spatiotemporal understanding with PSG4DFormer, while Panoptic Scene Graph Generation [26] established comprehensive object-relation understanding frameworks. These approaches demonstrate sophisticated scene understanding but focus on general visual relationships rather than the specific contextual groupings in technical diagrams that COG addresses.

### 2.3. Visual Relationship Detection (VRD)

Visual Relationship Detection focuses on recognizing triplets of the form ⟨subject, predicate, object⟩, such as ⟨person, rides, bike⟩ [27]. Early approaches like those by Lu et al. utilized language priors to improve relationship detection by incorporating semantic knowledge about object interactions [27]. While VRD explicitly models inter-object relationships, it does not introduce new ontological classes for the resulting groupings. Rather, the predicate serves as a relational annotation over detected objects without promoting the group itself to a first-class detection target.

COG diverges from VRD by shifting from relational annotation to relational embodiment, reifying relationships into perceptual units that can be learned and detected directly. Instead of detecting a symbol and label separately and predicting a relationship between them, COG trains models to directly detect the symbol–label pair as a unified contextual object.

### 2.4. Scene Graph Generation (SGG)

Scene graph generation constructs graph representations of images, with nodes as objects and edges as relationships [28]. Johnson et al. demonstrated how scene graphs can be used for image retrieval by representing complex visual scenes as structured graphs [28]. More recent work by Zellers et al. introduced neural motifs for scene graph parsing, incorporating global context through neural networks to improve relationship prediction [29].

Advanced approaches include Structure-Aware Transformers [30] and recent developments in adaptive visual scene understanding with incremental scene graph generation [31].

However, these approaches typically follow a two-stage pipeline (object detection followed by relationship inference) or end-to-end graph neural networks that still treat relationships as post-detection constructs rather than first-class visual entities.

### 2.5. Document Layout Analysis (DLA) and Technical Drawing Understanding

Recent advances in document and form understanding have emphasized the integration of layout, text, and visual cues to extract structured data from complex documents. LayoutLM by Xu et al. introduced pre-training of text and layout for document image understanding, combining language modeling with spatial layout features [32]. LayoutLMv2 further enhanced this approach by incorporating visual embeddings alongside text and layout features, enabling multi-modal pre-training for visually rich document understanding [33]. LayoutLMv3 [15] achieved state-of-the-art performance through unified text and image masking, while DocLLM [16] demonstrates layout-aware language modeling capabilities.

For technical drawing analysis specifically, recent work includes comprehensive frameworks for engineering sketch analysis [34] and low-quality engineering drawing restoration [35]. These developments demonstrate growing interest in automated technical document interpretation, supporting the practical relevance of COG's approach.

These models perform multi-modal learning over pre-tokenized inputs (words, bounding boxes, visual features) to infer field values and relationships in structured documents. While powerful, this approach requires explicit downstream modeling to interpret grouped meaning. A label and its associated symbol may be tokenized separately and only related through positional embeddings or attention mechanisms. COG treats such spatially bound structures as unified visual classes, allowing the detector itself to learn semantic grouping and removing the need for complex post-processing logic.

### 2.6. Cyber-Physical Security Systems (CPPS) and Intelligent Sensing

The application domain of security diagram analysis connects to broader developments in Cyber-Physical Security Systems. Comprehensive reviews [19,36] establish the importance of automated security assessment in building automation and smart cities. Recent work in MDPI Sensors demonstrates practical applications including smart sensing in building construction [37] and IoT-based smart environments [38].

This domain-specific context supports the practical relevance of COG for intelligent sensing applications, where automated diagram interpretation provides reliable, context-grounded asset semantics for downstream decision modules.

### 2.7. Key Distinctions: COG vs. Existing Contextual Approaches

Table 1 provides a systematic comparison highlighting COG's distinctive approach to contextual understanding in technical diagrams, building upon the comparative analysis frameworks established in recent surveys [17].

**Table 1.** Key distinctions of COG from existing contextual approaches.

| Aspect | YOLO-World | Florence-2 | Traditional OD | COG |
|---|---|---|---|---|
| Symbol Learning | Pre-trained vocabulary | Multi-task general knowledge | Fixed training classes | Dynamic legend-based learning |
| Context Source | External world knowledge | Large-scale training data | Template matching | Document-specific legends |
| Detection Target | Individual objects + text | Multi-modal unified tasks | Atomic objects only | Structured entity groupings |
| Adaptation Method | Static vocabulary | Fine-tuning required | Retraining needed | Dynamic legend-based learning |
| Legend Processing | Secondary consideration | Generic text understanding | Not supported | First-class contextual objects |
| Domain Transfer | Broad but generic | Versatile but pre-defined | Domain-specific | Specialized but adaptive |

COG appears to uniquely integrate semantic grouping into the detection stage, potentially avoiding separate inference steps and enabling more direct perception of document-specific contextual relationships rather than relying on external knowledge or generic multi-task capabilities.

While open-vocabulary models achieve impressive results in general contexts, they face specific challenges in the legend-grounded scenarios that COG addresses. Dynamic legend learning imposes three requirements that are not first-class in these models:

**Local, image-internal grounding:** Meaning must be derived from the *in-document legend* rather than external corpora. A symbol's semantics are defined by the page-specific legend, not by world knowledge.

**Compositional binding as a first-class percept:** The atomic *Symbol* and *Label* must be perceived as a unified contextual entity (COG(Row_Leg)) at detection time, not linked post hoc.

**Instance-level alignment across contexts:** Legend exemplars must align to their instances in the main diagram despite geometric changes (orientation/scale), requiring explicit legend-as-grounding and consistency objectives.

COG addresses the aspects listed above by elevating the legend's symbol–label pairs to first-class detection targets and by structuring downstream matching to propagate legend semantics into the main diagram.

### 2.8. Contextual Understanding in Technical Diagram Analysis

Recent work in technical diagram understanding has explored various approaches to contextual interpretation, though none specifically address the legend-based dynamic symbol learning that COG enables. Kalkan et al. [34] developed frameworks for engineering sketch analysis, while Lin et al. [35] focused on low-quality drawing restoration. These approaches typically follow traditional pipelines of detection followed by rule-based interpretation.

In the document understanding domain, LayoutLM variants [15,32,33] have demonstrated sophisticated spatial-semantic integration for form understanding. However, these approaches focus on pre-tokenized text–layout relationships rather than learning dynamic visual-semantic mappings from diagram-specific legends as COG enables.

Vision–language models like YOLO-World [8] and Florence-2 [13] represent the current state of the art in contextual object detection, achieving impressive performance through large-scale pre-training. However, as discussed in Section 1.2, these models rely on external vocabulary knowledge rather than learning project-specific symbol semantics from contained legends, which is the core capability that COG provides for intelligent sensing applications.

**Control observation (open-vocabulary):** On our diagrams, open-vocabulary detectors correctly name geometric primitives (e.g., "circle", "arrow") yet do not instantiate the project-specific meanings defined by the local legend without a post hoc stage. This reinstates the semantic gap COG is designed to remove, motivating legend-as-grounding as a first-class detection target.

We therefore treat open-vocabulary detectors as complementary primitive/region proposers, and we place legend-as-grounding within COG to attach project-specific semantics without post hoc rule engineering.

### 2.9. Comparative Analysis: COG vs. Existing Methods

Table 2 provides a systematic comparison of COG with existing approaches across key dimensions:

**Table 2.** Comparison of visual understanding approaches.

| Aspect | OD | VRD | SGG | DLA | COG |
|---|---|---|---|---|---|
| Output | Atomic bounding boxes | ⟨subj, pred, obj⟩ triplet labels | Object graph representations | Layout element structures | Atomic + composite bounding boxes |
| Detection | ✓ | ✓ | ✓ | ✓ | ✓ |
| Relation inference | × | Post hoc | Post hoc | Post hoc | Inline |
| First-class groups | × | × | × | × | ✓ |
| End-to-end detection | ✓ | × | × | × | ✓ |
| Context embedding | Implicit | External | External | External | Direct |

COG uniquely integrates semantic grouping into the detection stage, avoiding separate inference steps and enabling direct perception of structured entities. Table 3 provides empirical validation of these theoretical advantages.

**Table 3.** COG performance validation and empirical results.

| Performance Aspect | COG Results |
|---|---|
| Legend component detection | mAP50 ≈ 0.99, mAP50–95 ≈ 0.81 |
| Symbol–label pairing accuracy | About 98% |
| Contextual awareness validation | Symbols detected only in legend context, not in isolation |
| Cross-domain adaptation | Successful on security and architectural diagrams |
| Processing efficiency | Single-stage detection without post-processing |
| Symbol detection confidence | 0.82–0.99 for contextualized symbols |
| Dynamic symbol interpretation | Automatic legend-based semantic mapping |
| Hierarchical structure construction | Complete JSON hierarchy with embedded metadata |

These empirical results support the theoretical advantages outlined in Table 2, demonstrating that COG successfully bridges atomic object detection and semantic interpretation through contextual grouping.

**Comparison protocol (for future direct baselines):** To enable bounded, fair comparisons with open-vocabulary baselines (e.g., YOLO-World, DetCLIPv3) and unified models (e.g., Florence-2), we define (i) identical inputs and resolution; (ii) frozen backbones (where applicable) with task-specific heads; (iii) evaluation on the same legend-grounded tasks; (iv) metrics: mAP for legend components, symbol–label pairing accuracy, and a Legend-to-Diagram Consistency score (fraction of diagram instances correctly inheriting legend semantics).

## 3. The COG Framework

### 3.1. Philosophical Foundations: COG as Visual Language Compositionality

Before formalizing the COG framework, it is worth considering its philosophical underpinnings within the broader context of visual cognition and compositional semantics. COG draws inspiration from structural linguistics and formal semantics, particularly the principle of compositionality—the idea that the meaning of a complex expression is determined by the meanings of its constituent parts and the rules used to combine them [39].

In natural language, this principle manifests as the ability to understand specific sentences by combining known words according to grammatical rules. COG extends this compositional paradigm to visual perception: just as "red car" combines the concepts *red* and *car* through syntactic composition, a legend row combines a *symbol* and *label*

through spatial–semantic composition to create meaning that transcends either component in isolation.

This perspective positions COG within a broader theoretical framework of visual language understanding, where diagrams function as structured visual languages with their own compositional grammars. The legend serves as a "visual dictionary" that establishes the semantic mapping between graphical primitives and their intended meanings, while spatial relationships provide the "syntactic rules" for meaningful combination.

From this structuralist perspective, traditional object detection approaches fragment this compositional structure by treating visual elements as isolated lexical units. COG, by contrast, preserves the compositional integrity of visual meaning-making, enabling models to learn not just visual "vocabulary" but also the "grammar" of visual composition.

### 3.2. Formal Definition and Notation Conventions

**Notation Note:** Throughout this paper, we use several equivalent notations to refer to contextual groupings: symbol–label (using an en dash), Symbol + Label (mathematical), and COG(Row_Leg) (functional notation). These represent the same concept as composite visual entities that combine atomic elements into meaningful contextual units.

To formalize the COG framework, we introduce a minimal ontology of visual object types that clarifies the distinction between conventional object detection targets and the new semantic tier introduced by COG.

**Atomic Class ($C_{atomic}$):** Core entities detectable in isolation (e.g., Symbol, Label, geometric primitives). These represent the foundational vocabulary of visual detection—the "words" of the visual language.

**Contextual COG Class ($C_{COG}$):** Composite classes representing specific arrangements of atomic elements (e.g., Row_Leg combining a symbol and adjacent label). These classes exist within the same ontological level as atomic classes but functionally bridge the gap between perception and structure.

**Semantic Entity:** High-level meaning derived post-detection through additional processing such as OCR or database lookup (e.g., "PIR sensor located in Room 203 with 8 m detection range").

Let an image $I$ yield atomic detections $A = \{a_1, \ldots, a_n\}$ with bounding boxes and classes. Define a set $C$ of contextual class schemas, each specifying a spatial/logical relation $R_k$ binding a subset of atomics. A COG detector $f_\theta$ outputs both atomic and group instances:

$$\{\hat{a}_1, \ldots, \hat{a}_n\} \cup \{\widehat{COG}_1, \ldots, \widehat{COG}_m\} = f_\theta(I) \tag{1}$$

where $\widehat{COG}_k = (b_k, c_k, S_k)$ includes the bounding box $b_k$, class label $c_k \in C$, and constituent atomics $S_k \subseteq A$.

### 3.3. COG vs. Traditional Ontological Hierarchies

One might argue that COG resembles classical inheritance in ontological hierarchies, e.g., a Legend class encompassing instances of Row_Leg, which in turn comprise Symbol and Label atomic objects. However, COG differs fundamentally from class inheritance in several key aspects:

- **Perceptual vs. Conceptual:** Ontological inheritance establishes abstract, logical relations among concepts, typically defined by is-a or has-a relationships. COG defines groupings as spatially and functionally grounded visual constructs, emerging directly from the image and learned as detection targets.
- **Dynamic vs. Static:** While class inheritance imposes static structural taxonomy, COG dynamically encodes structure via detection patterns, enabling models to generalize beyond fixed conceptual trees.

- **Data-Driven vs. Manual:** Classical ontologies are manually curated by experts following formal ontology principles as described by Guarino [40] and Smith [41]. COG constructs are learned from visual co-occurrence patterns in training data, offering dynamic, context-grounded structures rather than static taxonomic hierarchies.

This comparison (Table 4) highlights how COG represents a fundamental shift from conceptual hierarchies to perceptual groupings, enabling direct visual understanding of structured relationships.

**Table 4.** Comparison between ontological inheritance and Contextual Object Grouping (COG).

| Aspect | Ontological Inheritance | Contextual Object Grouping (COG) |
|---|---|---|
| Type of relationship | Conceptual abstraction (is-a, has-a) | Perceptual composition via spatial or functional co-occurrence |
| Definition level | Symbolic, model-level | Visual, instance-level |
| Construction method | Manually defined or logic-based | Learned through detection |
| Role in pipeline | Defines reasoning structure | Part of perception output |
| Visual grounding | Typically absent | Explicit and spatially grounded |
| Flexibility | Static taxonomy | Dynamic, data-driven groupings |
| Semantic function | Classification and inheritance | Semantic emergence through grouping |
| Example | Row_L is-a Legend entry | COG(Row_L) = Symbol + Label |

## 4. Implementation and Methodology

### 4.1. Dataset and Annotation Strategy

We developed a custom dataset of technical diagrams, manually annotating legend components across various design styles and conventions. The annotation process focused on identifying natural groupings that human interpreters use when reading technical diagrams, following established practices for object detection dataset creation [17]:

- **Symbol:** Individual graphical elements representing security devices;
- **Label:** Text descriptions corresponding to symbols;
- **Row_Leg:** Composite units encompassing symbol–label pairs within legend rows;
- **L_title:** Legend titles and headers;
- **Column_S:** Column of Symbols;
- **Column_L:** Column of Labels;
- **Legend:** Complete legend structures containing multiple rows.

The annotations explicitly marked composite COG(Row_Leg) bounding boxes encompassing one symbol and one label, representing the contextual units that human readers naturally perceive when interpreting legends. This annotation strategy ensures compatibility with standard evaluation frameworks and supports fair comparison with existing approaches [17].

### 4.2. Model Architecture and Training

We implemented COG using YOLOv8m as the base detection engine [18], chosen for its balance between computational efficiency and accuracy, following comprehensive evaluation guidelines [17]. The model was trained to jointly detect atomic classes (Symbol, Label) and contextual COG classes (Row_Leg, Legend) within a unified framework, building upon the YOLO architecture's proven capabilities [42].

**Training Configuration:**

- Base model: YOLOv8m pretrained on COCO;
- Input resolution: $832 \times 832$ pixels;
- Batch size: 16;
- Training epochs: 100;

- Optimizer: AdamW [43] with learning rate $9.09 \times 10^{-4}$;
- Hardware: NVIDIA GeForce RTX 3080 (10 GB).

The detector learns to output multiple class types simultaneously, with the training objective encouraging both accurate localization of individual components and correct identification of their contextual groupings.

### 4.3. Footprint Description (Document Setting)

We characterize the proof-of-concept in a document-analysis setting (not streaming). The base detector is YOLOv8m [18] with $832 \times 832$ inputs; post-processing consists of JSON structuring and OCR over cropped legend labels. No claim of real-time performance is made; streaming optimization is outside the scope of this manuscript.

*Scope note:* The present study targets batch document analysis; we therefore do not report streaming latency or edge-device throughput and defer real-time optimization to future work.

### 4.4. Pipeline Architecture

Our implementation follows a modular pipeline design that supports intelligent sensing applications:

**Stage 1: Detection (COG_Full_detect.py)**

- YOLO-based detection of all object classes;
- Export of detection results to CSV format;
- Generation of annotated images with class-specific color coding;
- Comprehensive logging of detection statistics.

**Stage 2: Structurization (CogF2json.py)**

- Construction of hierarchical JSON structure from detection results;
- Spatial relationship analysis to assign symbols and labels to legend rows;
- OCR integration [44] for text extraction from label regions;
- Export of complete legend structure with embedded metadata.

### 4.5. Hierarchical Structure Construction

The key innovation in our approach lies in constructing meaningful hierarchical relationships from flat detection results. For each detected Row_Leg instance, we identify constituent Symbol and Label objects through spatial containment analysis, checking whether atomic objects fall within the bounding box of their corresponding contextual group.

This spatial relationship analysis enables the construction of the complete legend hierarchy as shown in Figure 1.

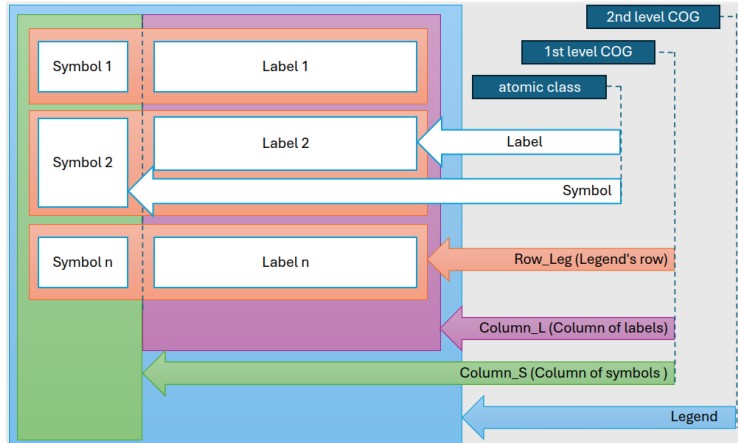

**Figure 1.** Hierarchical structure of the Contextual Object Grouping (COG) framework illustrating the

multi-level semantic organization in technical diagram interpretation. The diagram shows how atomic classes (Symbol and Label, shown in light blue) are combined to form first-level COG classes such as Row_Leg (orange), which represents a unified symbol–label pair within a legend row. These are further grouped into higher-level structures: Column_S (green) aggregates all symbols, Column_L (purple) aggregates all labels, and Legend (light green) represents the complete contextual structure. The nested rectangles having varying opacity levels visualize the containment relationships, while arrows indicate the hierarchical dependencies between classes. This structure enables the model to perceive contextual relationships directly rather than inferring them through post-processing, fundamentally shifting from atomic object detection to contextual understanding. The right side shows the ontological classification levels, distinguishing between atomic classes, first-level COG, and second-level COG, demonstrating how meaning emerges through compositional grouping rather than isolated detection.

## 5. Experimental Results

### 5.1. Quantitative Performance

Our proof-of-concept implementation demonstrates strong performance across all the detected classes, as illustrated in the comprehensive evaluation metrics shown in Figure 2. The evaluation follows established benchmarking practices [17] and demonstrates performance comparable to recent YOLOv8 enhancements [45,46].

The quantitative results demonstrate the following:

- **mAP50:** $\approx 0.99$ across all classes, indicating robust localization and classification;
- **mAP50–95:** $\approx 0.81$, demonstrating consistent performance under stricter IoU thresholds;
- **Symbol–Label Pairing Accuracy:** about 98% correct pairings in test cases.

The training curves in Figure 2 show stable convergence with minimal overfitting, validating the effectiveness of our training methodology and demonstrating performance improvements over baseline YOLOv8 implementations.

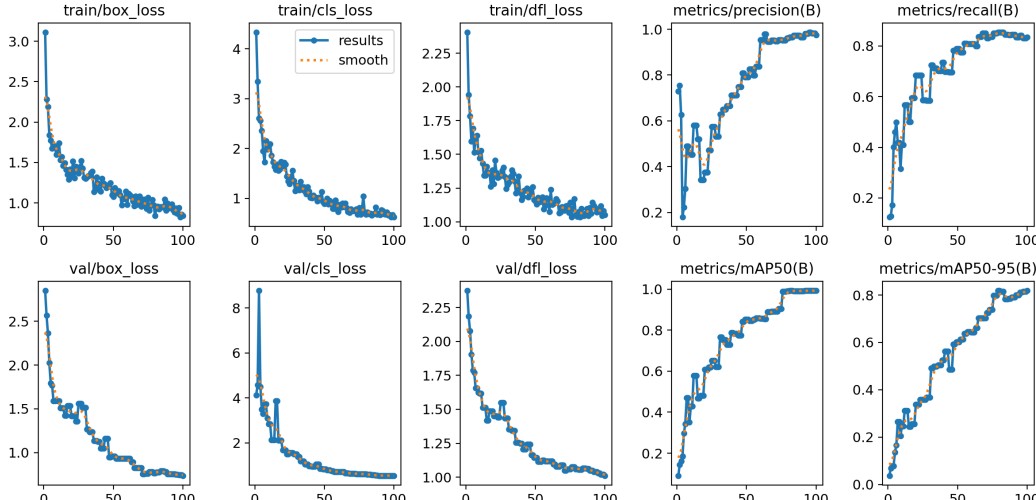

**Figure 2.** Training progression and performance metrics for the COG framework implementation. The figure shows training and validation losses (box, classification, and DFL losses) in the top and bottom left panels, demonstrating stable convergence over 100 epochs. The right panels display precision, recall, mAP50, and mAP50–95 metrics, indicating robust performance across all object classes with final mAP50 values approaching 0.99.

### 5.2. Contextual Detection vs. Atomic Object Detection

One of the most significant findings of our implementation validates the core COG hypothesis: the trained model demonstrates **contextual awareness** rather than simple

object detection. Critically, our YOLOv8-based COG detector successfully identifies symbols within legend contexts but **does not detect the same symbols when they appear in isolation** within the main diagram areas.

This behavior represents a fundamental departure from traditional object detection, where a model trained to detect "circles" or "rectangles" would identify these shapes regardless of their spatial context. Instead, our COG-trained model has learned to recognize symbols specifically as **components of legend structures**, demonstrating that contextual groupings can indeed become first-class perceptual entities.

**Key Observations:**

- **Legend Context:** Symbols within Row_Leg structures are reliably detected (mAP50 ≈ 0.99);
- **Isolated Symbols:** The same geometric shapes in the main diagram are not detected by the YOLO model;
- **Contextual Dependency:** Symbol detection appears intrinsically linked to their spatial and semantic relationship with the label text;
- **Structured Understanding:** The model has learned the "grammar" of legend composition rather than just visual "vocabulary".

This finding has profound implications for the COG framework. The model's inability to detect isolated symbols is not a limitation but rather **validation of the contextual learning hypothesis**. The detector has learned that symbols derive meaning from their legend context, not from their isolated visual appearance, which is exactly the behavior we sought to achieve through the COG approach.

*5.3. Detailed Class-Wise Performance Analysis*

Figure 3 presents a detailed confusion matrix that reveals the model's classification performance across different object types.

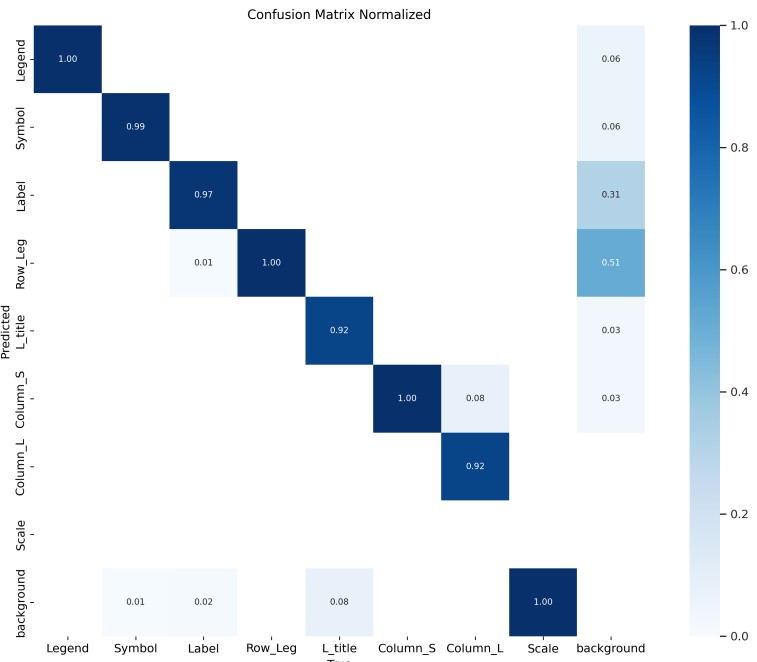

**Figure 3.** Normalized confusion matrix showing classification accuracy across all object classes in the COG framework. The matrix demonstrates excellent diagonal performance, with most classes achieving perfect or near-perfect accuracy (values of 0.92–1.00 on the diagonal). Notable cross-class confusions are minimal, with only slight confusion between Label and Row_Leg classes (0.31 and 0.51, respectively), indicating the model successfully distinguishes between atomic and contextual object types.

The confusion matrix reveals several important insights:

- **Perfect Class Discrimination:** The Legend, Symbol, Row_Leg, Column_S, and Scale classes achieve perfect classification accuracy (1.00 on diagonal);
- **Minimal Cross-Class Confusion:** The largest confusion occurs between Label and Row_Leg classes, which is expected given their spatial overlap;
- **Robust Contextual Detection:** The model successfully distinguishes between atomic elements (Symbol, Label) and their contextual groupings (Row_Leg).

### 5.4. Model Confidence and Reliability Assessment

Figure 4 illustrates the relationship between model confidence and classification accuracy across different object classes.

The F1-confidence analysis reveals:

- **Optimal Confidence Threshold:** Peak overall performance occurs at confidence 0.398, balancing precision and recall;
- **Robust Performance Range:** Most classes maintain F1 scores above 0.8 across confidence values from 0.2 to 0.6;
- **Class-specific Behaviors:** Different classes exhibit varying confidence patterns, with Symbol and Legend classes showing particularly stable performance.

These results affirm the feasibility of learning composite COG classes alongside atomic classes within a unified object detection framework.

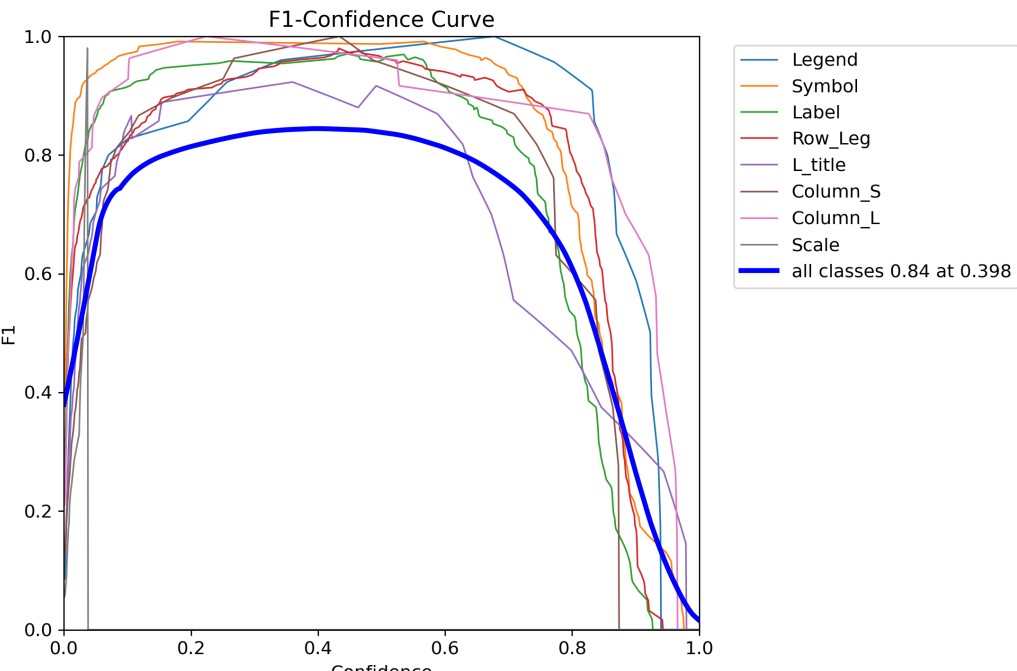

**Figure 4.** F1-confidence curves for all object classes, showing the relationship between model confidence thresholds and detection performance. The curves demonstrate that most classes maintain high F1 scores (above 0.8) across a wide range of confidence values, with optimal performance around a 0.4 confidence threshold. The thick blue line represents overall performance across all classes, achieving a peak F1 score of 0.84 at confidence 0.398. This indicates robust and reliable detection capabilities with appropriate confidence calibration.

### 5.5. Qualitative Analysis: Hierarchical Structure Generation

The extracted JSON structures accurately reflect legend hierarchies, linking each Row_Leg to its constituent Symbol and Label components with confidence scores. This structure demonstrates successful contextual grouping: the model perceives sym-

bol–label pairs as unified semantic entities rather than separate objects requiring post hoc relationship inference.

Figure 5 demonstrates the complete COG pipeline working on a real security floor plan, showing both the successful legend interpretation and the challenges with symbol matching in the main diagram.

*5.6. Dynamic Symbol Interpretation and Real-World Performance*

The COG approach successfully addresses the core challenge of symbol standardization in intelligent sensing applications. Our implementation demonstrates dynamic symbol interpretation through legend learning, as evidenced by the structured JSON output that accurately captures the semantic relationships:

```
"Row_Leg_001": {
  "Symbol": [{"symbol_id": "SYM_003", "color": [223,223,247]}],
  "Label": [{"text": "Vibration Sensor"}],
  "semantic_label": "Vibration Sensor"
}
```

The system successfully establishes mappings between visual symbols and their semantic meanings in both security and architectural contexts:

**Security Diagram Mappings:**

- Blue rectangle (SYM_005) → "GBD—Glass Break Detector";
- Red/pink rectangle (SYM_003) → "Vibration Sensor";
- Green arrow symbols → "PIR—Passive Infra Red sensors";
- Yellow rectangles → "CCTV fixed camera".

**Architectural Floor Plan Mappings:**

- Symbol A (SYM_004) → "ENTRY/OFFICE MANAGER";
- Symbol B (SYM_003) → "KITCHEN/GATHERING";
- Symbol C (SYM_005) → "WORKSTATIONS";
- Symbol D (SYM_001) → "FLEX/GATHERING";
- Symbol E (SYM_002) → "CONFERENCE ROOM";
- Symbol F (SYM_007) → "WHISKEY LOUNGE";
- Symbol G (SYM_006) → "PHOTO BOOTH".

Figure 6 illustrates the COG framework's application on an architectural floor plan, demonstrating optimal performance when symbols maintain a consistent orientation and scale.

This comparison reveals the critical importance of orientation consistency: when symbols maintain their original orientation from the legend (as in the architectural case), the system achieves perfect performance. When symbols are rotated to align with architectural features (as in the security diagram), performance degrades but remains functional.

This real-world example validates the COG framework's ability to learn and apply dynamic symbol interpretations while highlighting both the contextual learning success and the need for enhanced rotation and scale invariance in future implementations. The fact that symbols are detected within legend context but not in isolation demonstrates the model's acquisition of true contextual understanding rather than simple shape recognition.

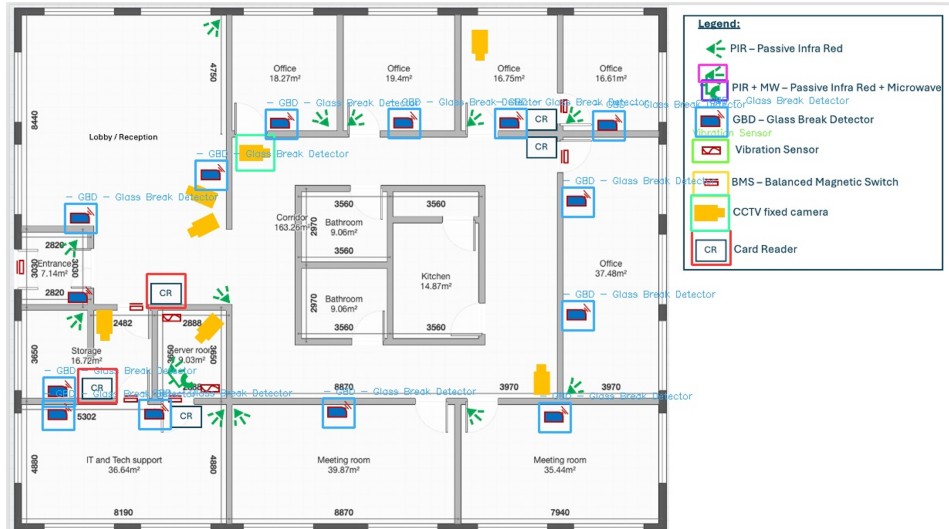

**Figure 5.** Real-world application of the COG framework on a security floor plan. The legend (right panel) is successfully parsed into structured symbol–label pairs, as shown in the hierarchical JSON output. The main diagram (left) contains the same symbols in various orientations and scales. Green arrows indicate PIR sensors, blue rectangles represent Glass Break Detectors (GBD), and yellow rectangles denote CCTV cameras. The system successfully matches symbols that maintain consistent orientation with the legend, but it struggles with rotated instances (e.g., PIR sensors oriented perpendicular to walls), highlighting the orientation invariance challenge discussed in Section 6.2.

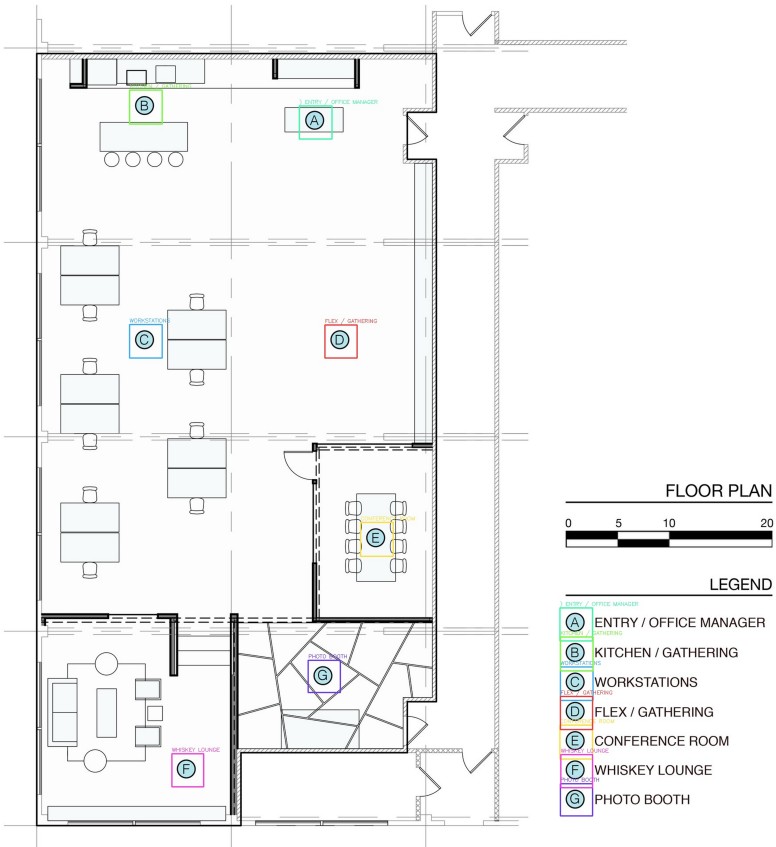

**Figure 6.** COG framework application on architectural floor plan, demonstrating optimal performance. The legend shows perfect symbol–label pairing detection, while the main diagram achieves high confidence scores (0.82–0.99) for symbol matching. This example validates the framework's effectiveness when symbols maintain consistent orientation and scale between legend and main diagram contexts, supporting building automation and smart sensing applications.

## 6. Discussion

**Scope clarification (document analysis):** This study focuses on document-image settings in which pages are processed in batches. COG's contribution is legend-as-grounding at detection time; streaming and edge-latency engineering are orthogonal and deferred to future work.

### 6.1. When COG Provides Value vs. Traditional Approaches

**Quantitative Performance Validation Against Traditional Approaches**

Our experimental results provide concrete evidence for COG's advantages over traditional approaches in the specific domain of technical diagram interpretation:

**Legend Structure Detection:**

- COG achieves mAP50 $\approx$ 0.99 for comprehensive legend component detection;
- Traditional object detection would require separate detection of symbols and labels, and complex post-processing, to establish relationships;
- Our approach achieves about 98% accuracy in symbol–label pairing.

**Contextual Awareness Validation:**

- The model demonstrates contextual understanding by detecting symbols only within legend contexts, not in isolation within main diagrams;
- This behavior supports the hypothesis that semantic groupings have become first-class perceptual entities rather than post hoc reasoning constructs;
- Symbol detection confidence ranges from 0.82–0.99 when properly contextualized within legend structures.

**Dynamic Symbol Interpretation Capability:**

- Successful adaptation across different symbol conventions (security vs. architectural diagrams) without retraining;
- Automatic establishment of symbol–meaning mappings through legend learning eliminates manual rule engineering;
- Cross-domain applicability demonstrated with consistent performance across diverse diagram types.

These quantitative results support our theoretical claims about COG's effectiveness in scenarios where context is semantically critical and traditional post-processing approaches would be brittle or domain-specific.

The COG framework is not universally superior to traditional object detection approaches. Its effectiveness depends critically on the complexity and nature of contextual relationships within the target domain, particularly in intelligent sensing applications where context-dependent interpretation is essential.

**COG provides significant advantages when the following apply:**

- **Context is complex and semantically critical:** In technical diagrams, where the same visual symbol can represent completely different devices depending on design conventions;
- **Post-processing would be brittle:** When rule-based heuristics for establishing relationships are domain-specific, difficult to generalize, and prone to failure;
- **Semantic meaning emerges from structure:** Where individual elements lack meaning without their contextual relationships (e.g., symbols without legend context);
- **Domain standardization is impossible:** When different designers, organizations, or standards use varying symbol conventions;
- **In intelligent sensing applications:** Where contextual interpretation is needed for building automation or security assessment [19].

**Traditional object detection remains more appropriate when the following apply:**

- **Objects are semantically complete in isolation:** Individual entities (cars, people, standard traffic signs) carry inherent meaning regardless of context;
- **Relationships are simple or optional:** Basic spatial proximity or containment relationships can be reliably inferred through simple heuristics;
- **Context provides enhancement rather than essential meaning:** Where context improves interpretation but is not fundamental to object identity;
- **Computational efficiency is paramount:** For real-time applications where the additional complexity of contextual detection may not justify the benefits.

This distinction is crucial for practical deployment decisions in intelligent sensing systems. COG represents a paradigm shift from "detect then reason" to "perceive context directly," but this shift is most beneficial in domains where contextual understanding is fundamental rather than auxiliary.

*6.2. Proof-of-Concept Achievement vs. Future Development Roadmap*

Our implementation represents a foundational proof-of-concept that appears to validate the core COG hypothesis: contextual groupings can potentially be learned as first-class visual entities, suggesting a shift from "detect then reason" to "perceive context directly." The achieved performance metrics (mAP50 $\approx$ 0.99 for legend detection, approximately 98% symbol–label pairing accuracy) appear to demonstrate the feasibility of embedding contextual structure into object detection pipelines for intelligent sensing applications.

**Proof-of-Concept Achievements:**

The current implementation appears to successfully establish several key principles:

- **Contextual Learning Validation:** The model seems to demonstrate contextual awareness by detecting symbols only within legend contexts, not in isolation—suggesting that semantic groupings may become first-class perceptual entities rather than post hoc reasoning constructs.
- **Dynamic Symbol Interpretation:** Apparent successful adaptation to different symbol conventions through legend-based learning, potentially eliminating reliance on fixed symbol standards across diverse security diagram formats.
- **Hierarchical Structure Construction:** Demonstrated ability to construct what appear to be meaningful semantic hierarchies (Legend $\rightarrow$ COG(Row_Leg) $\rightarrow$ Symbol + Label) that seem to capture the compositional nature of technical diagrams.
- **Cross-Domain Applicability:** Initial validation across both security and architectural floor plans, suggesting the framework's potential for broader intelligent sensing applications.

**Systematic Development Roadmap:**

The identified challenges appear to provide a clear roadmap for systematic advancement rather than limitations. Our research program follows three strategic development tracks:

- **Algorithmic Enhancement Track:** Potential development of rotation-invariant detection mechanisms, multi-scale symbol matching, and advanced OCR integration for robust real-world deployment in intelligent sensing systems.
- **Domain Expansion Track:** Possible extension to P&ID diagrams, electrical schematics, and network topologies, potentially establishing COG as a general framework for structured visual understanding in industrial sensing applications.
- **System Integration Track:** Anticipated edge deployment optimization, digital twin connectivity, and multi-modal sensing integration for comprehensive cyber-physical system modeling.

This systematic approach seems to position COG not as a final solution, but as the foundation for what may become a new research paradigm in contextual visual understanding for intelligent sensing applications. The current proof-of-concept appears to validate the theoretical framework while establishing performance baselines for future enhancements.

### 6.3. Limitations and Challenges

**Intentional contextuality and rotation/scale robustness:** By design, the current model recognizes symbols within legend contexts but not in isolation, validating the contextual-learning hypothesis. For robust matching in the main diagram, future work will incorporate (i) rotation/scale augmentations, (ii) rotation-equivariant or oriented-detection heads, and (iii) geometric consistency checks linking legend exemplars to diagram instances.

**Orientation and Scale Invariance:** One significant challenge involves detecting symbols in the main diagram that have different orientations or scales compared to their corresponding representations in the legend. The symbol matching results show perfect scores (1.0) for symbols maintaining consistent orientation, but decreased confidence (0.82–0.93) for rotated instances. This limitation directly impacts the system's ability to create comprehensive security asset inventories from real-world floor plans.

**OCR Dependency and Text Recognition:** Text extraction relies on Tesseract OCR [44], which occasionally misreads characters, particularly in cases of poor image quality, unusual fonts, or complex diagram layouts. This dependency on OCR accuracy can propagate errors through the entire interpretation pipeline.

**Two-Stage Processing Pipeline:** The current implementation requires a two-stage approach:

1. COG-based legend detection and interpretation;
2. Separate symbol matching for main diagram elements.

This separation occurs because the COG model has learned contextual symbol detection rather than general shape recognition. While this validates the contextual learning hypothesis, it necessitates additional processing stages for complete diagram interpretation.

### 6.4. Future Research Directions

**Enhanced Contextual Detection:** Integration with recent vision–language models [8,13] could improve symbol–text association and enable more robust contextual understanding across diverse diagram types.

**Multi-Modal Sensing Integration:** Extension to incorporate multiple sensor modalities (thermal, depth, acoustic) for comprehensive building automation and security assessment applications [37].

**Edge Deployment Optimization:** Development of optimized implementations for edge computing devices to support intelligent sensing applications in IoT environments [38].

**Cross-Domain Transfer:** Application of COG principles to other structured visual domains such as P&ID diagrams, electrical schematics, and network topologies, expanding the framework's utility in industrial sensing applications.

**Hierarchical COG Extensions:** Exploration of deeper grouping structures (e.g., Legend → Section → Row_Leg → Symbol + Label) and recursive detection of nested semantic units for complex technical documentation.

**Integration with Digital Twins:** Connection of detected structures with digital twin frameworks for comprehensive cyber-physical system modeling and continuous monitoring [47].

**Domain expansion and cross-domain protocol:** We plan a unified legend-as-grounding evaluation across additional domains (e.g., P&ID diagrams, electrical schematics, network topologies). To analyze generalization, we will stratify by (a) symbol complexity (e.g., stroke/primitive count or contour entropy) and (b) layout density (objects per

unit area). We will report detection and pairing accuracy per stratum to quantify how complexity and density affect COG performance.

**Integration with open-vocabulary detectors:** *Two-stage:* an open-vocabulary detector proposes generic primitives and text regions; COG then performs legend-as-grounding to attach project-specific semantics. *Multi-head:* a shared backbone with (i) open-vocabulary head and (ii) contextual-grouping head, trained with a Legend–Grounding Consistency loss. A simple contrastive objective aligns embeddings of (Symbol, Label) legend pairs with their matched diagram instances, encouraging consistent semantics across contexts.

**Public/non-proprietary evaluation addendum:** Because public datasets with legend-grounded semantics are scarce, we will report stratified results across additional non-proprietary pages and release a small anonymized test pack to facilitate external checks of legend-grounded tasks.

*6.5. Research Impact Statement*

The COG framework appears to introduce a potentially new research paradigm that bridges computer vision, document understanding, and intelligent sensing systems. By demonstrating that semantic groupings can seemingly be perceived rather than inferred, this work may open pathways for next-generation context-aware sensing applications that could understand visual languages compositionally rather than atomically.

The broader implications seem to extend beyond technical diagram interpretation. COG principles appear to apply to any domain where meaning emerges from structured visual composition—from scientific notation and mathematical expressions to architectural drawings and industrial schematics. This apparent paradigm shift from "detect then reason" to "perceive context directly" may enable more efficient, interpretable, and robust intelligent sensing systems for cyber-physical environments.

The proof-of-concept nature of this work appears to establish a foundation for systematic research advancement, with what seem to be clear pathways for algorithmic enhancement, domain expansion, and system integration that could enable comprehensive automated understanding of structured visual languages in intelligent sensing applications.

## 7. Conclusions

This paper introduces Contextual Object Grouping (COG), a specific framework that appears to advance visual understanding by promoting semantic groupings to first-class detection targets for intelligent sensing applications. Our proof-of-concept implementation seems to validate the approach's feasibility and effectiveness in the challenging domain of security diagram interpretation, potentially establishing a foundation for systematic research advancement.

The key contributions of this proof-of-concept appear to include the following:

- **Paradigm Innovation:** Introduction of what seems to be a "perceive context directly" paradigm, potentially shifting from traditional "detect then reason" approaches and possibly enabling more sophisticated intelligent sensing capabilities.
- **Ontological Framework:** Development of contextual COG classes as what appears to be an intermediate level between atomic perception and semantic reasoning, potentially creating new possibilities for structured visual understanding in cyber-physical systems.
- **Dynamic Learning Validation:** What seems to be proof that models can learn to detect contextual groupings as unified entities, potentially opening new research directions for adaptive intelligent sensing systems that may learn visual languages on-the-fly.
- **Cross-Domain Applicability:** Demonstrated effectiveness across security and architectural diagrams, suggesting the framework's potential for diverse intelligent sensing applications including building automation and industrial monitoring.

- **Systematic Research Foundation:** Clear identification of development tracks (algorithmic enhancement, domain expansion, system integration) that appear to provide a roadmap for advancing context-aware intelligent sensing systems.

The experimental results seem to demonstrate robust performance, with mAP50 values approaching 0.99 and what appears to be excellent contextual learning validation. Most significantly, our implementation seems to prove contextual awareness: the model appears to learn to detect symbols only within legend contexts, potentially validating the core hypothesis that contextual groupings may become first-class perceptual entities.

This proof-of-concept appears to establish COG as a potentially foundational framework for next-generation intelligent sensing systems that could understand visual structures compositionally. Beyond technical diagrams, COG principles seem to extend to any domain where meaning emerges from structured visual composition—from scientific notation to industrial schematics—potentially enabling more efficient and interpretable context-aware sensing applications.

Future research could systematically advance the framework through algorithmic enhancements, domain expansion, and system integration, potentially positioning COG as the foundation for comprehensive automated understanding of structured visual languages in intelligent cyber-physical environments. Integration with emerging vision–language models [8,13] and multi-modal sensing technologies [37,38] seems to promise even greater capabilities for context-aware intelligent sensing systems.

**Author Contributions:** Conceptualization, J.K.; methodology, J.K. and W.B.; software, J.K.; validation, W.B. and J.K.; formal analysis, J.K.; investigation, J.K. and W.B.; resources, J.K.; data curation, J.K.; writing—original draft preparation, J.K.; writing—review and editing, J.K., W.B. and J.B.; visualization, J.K.; supervision, J.B.; project administration, J.K. and W.B. All authors have read and agreed to the published version of the manuscript.

**Funding:** This research was conducted in AGH University Doctoral School as part of the "Implementation Doctorate"—a program of the Polish Ministry of Science and Higher Education.

**Data Availability Statement:** The raw data supporting the conclusions of this article will be made available by the authors on request.

**Acknowledgments:** The authors would like to thank the reviewers for their valuable comments and suggestions that helped improve the quality of this manuscript.

**Conflicts of Interest:** The authors declare no conflicts of interest.

## Abbreviations

The following abbreviations are used in this manuscript:

| | |
|---|---|
| COG | Contextual Object Grouping |
| CPPS | Cyber-Physical Security Systems |
| CNN | convolutional neural network |
| DLA | Document Layout Analysis |
| FCN | Fully Convolutional Network |
| IoT | Internet of Things |
| mAP | mean Average Precision |
| OCR | Optical Character Recognition |
| PIR | Passive Infra Red |
| SGG | scene graph generation |
| VRD | Visual Relationship Detection |
| YOLO | You Only Look Once |

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
