# Peer review of "Contextual Object Grouping (COG): A Specialized Framework for Dynamic Symbol Interpretation in Technical Security Diagrams"

_algorithms, doi:10.3390/a18100642_

Round 1
Reviewer 1 Report
Comments and Suggestions for Authors
1. The paper introduces a novel approach by embedding contextual understanding into the detection process through the COG framework. However, a comparison table with existing methods like YOLO-World and Florence-2 would help clarify the unique strengths of COG for readers.
2. While results on security diagrams are strong, especially with high mAP scores, the claim of cross-domain generalizability would be better supported by including tests on diagrams from other fields like electrical or architectural plans.
3. The introduction of a contextual COG class is promising, but the paper should address how the framework handles variable or incomplete legends. A brief note on integration with real-world tools like CAD or BIM would add practical relevance.
None
Author Response
Comment 1: "A comparison table with existing methods like YOLO-World and Florence-2 would help clarify the unique strengths of COG for readers."
Response: We have added Table 1 in Section 2.7 "Key Distinctions: COG vs. Existing Contextual Approaches" that provides a systematic comparison of COG with YOLO-World, Florence-2, and traditional object detection approaches. This table highlights COG's unique approach to dynamic legend-based learning and document-specific contextual understanding, distinguishing it from existing methods that rely on pre-trained vocabularies or generic multi-task capabilities.
Comment 2: "The claim of cross-domain generalizability would be better supported by including tests on diagrams from other fields like electrical or architectural plans."
Response: We appreciate this suggestion and have strengthened our discussion of cross-domain applicability. In Section 5.6, we demonstrate COG's effectiveness on both security and architectural floor plans, showing successful symbol-label mappings across different diagram types. Additionally, in Section 6.4 "Future Research Directions," we outline plans for extending COG to P&ID diagrams, electrical schematics, and network topologies as part of our systematic research roadmap.
Comment 3: "The paper should address how the framework handles variable or incomplete legends. A brief note on integration with real-world tools like CAD or BIM would add practical relevance."
Response: We have addressed these concerns in multiple sections:
Section 6.3 "Limitations and Challenges" now discusses variable legend configurations and processing pipeline considerations
Section 6.4 includes "Integration with Digital Twins" as a future research direction for comprehensive cyber-physical system modeling
Section 6.5 "Research Impact Statement" discusses broader implications for real-world integration with existing industrial tools
Reviewer 2 Report
Comments and Suggestions for Authors 1. The Fundamental Question Addressed by the Article: This article describes how symbols used in technical security diagrams acquire contextual meaning and how this contextual interpretation can be automated. The authors recommend using the "COG" (Contextual Object Grouping) approach, a new object recognition framework that recognizes the symbol and its description as a whole. 2. Originality and Relevance to the Field: The topic is both original and the authors have made original contributions to the article. Furthermore, the topics and information discussed are directly related to current research areas. The authors' solution has been applied to a problem where existing object recognition methods are inadequate, and a clear solution has been used to address the symbol standardization problem encountered in technical diagram interpretation. 3. Contribution to the Field The COG approach is recommended as a model that perceives symbols contextually rather than merely formally, recognizing the symbolic text within the legend as a unit of meaning. It can also be used in a variety of fields, including security systems, architectural plans, and industrial drawings. 4. Method and Criticism A concrete and applicable YOLOV8-based solution has been presented. Furthermore, the MAP50≈ 0.99 performance is quite remarkable. The hierarchical structure is modeled with graphical support. I have no criticisms of the article. The article's content is adequate and well-prepared. 5. Consistency of Results The results of the paper are extremely convincing and provide solid support for the presented hypothesis. 6. Sources and Figures The resources are sufficient and up-to-date. The graphics and figures used are clear, understandable, and explanatory. General Evaluation The content, grammar, and references used in the article are quite strong and sufficient, and I find it appropriate for the article to be published directly.Author Response
Response: We appreciate your positive evaluation and comprehensive review. Your recognition of COG's originality and practical applicability is encouraging. We have incorporated your feedback by:
Emphasizing the proof-of-concept nature throughout the manuscript
Strengthening the discussion of COG's potential for cross-domain applications
Maintaining the technical rigor while clearly positioning this as foundational research
We are pleased that you found the MAP50≈0.99 performance remarkable and the hierarchical structure modeling convincing. Your recommendation for publication motivates us to continue developing this research direction.
Reviewer 3 Report
Comments and Suggestions for Authors
The paper presents a framework named Contextual Object Grouping (COG) that structures visual understanding, particularly in technical diagram interpretation and cyber-physical security applications. However, the paper is not explicitly highlighting the research gap and urgency in the domain.
The authors need to clearly articulate the limitations of existing object detection and scene understanding methods in handling structured diagrams and complex contextual relationships.
The authors should explicitly describe how COG uniquely combines hierarchical ontological modeling with perceptual detection to achieve true contextual understanding. Highlight how this integration addresses challenges unresolvable by traditional methods, thus positioning the work as a significant leap forward.
Authors need to elaborate on more detailed descriptions of modifications made to YOLOv8 or other core models, including architecture changes, loss functions, and training procedures.
The formal equation effectively captures the detection of atomic and grouped entities but would benefit from clearer definitions of components and explicit representation of relational and contextual constraints for improved clarity and rigor, it is also important to include a citation for the specific formula used to ensure proper attribution and context
In the evaluation section, compare the COG with existing methods designed for understanding diagrams, detecting based on ontologies, or using language models to show how COG is better than traditional object detection, relation inference, or combined methods.
To improve clarity and impact, consider explicitly summarizing the key advantages of COG over traditional methods, such as its ability to perceive semantic groups directly and reduce reliance on post-processing, and better articulate potential limitations or areas for future work, especially regarding scalability, robustness to visual variations, and real-time implementation challenges.
Discussing how the current reliance on YOLOv8 influences the generalizability, scalability, and potential limitations of the framework would provide a more comprehensive perspective.
Author Response
Comment 1: "The paper is not explicitly highlighting the research gap and urgency in the domain."
Response: We have strengthened the research gap discussion. In Section 1.2, we added a detailed paragraph (lines 119-129) that articulates how advanced models like YOLO-World and Florence-2, despite their capabilities, cannot dynamically learn project-specific symbol semantics from contained legends. This positions COG as addressing a gap in contextual understanding for technical diagrams.
Comment 2: "Authors should explicitly describe how COG uniquely combines hierarchical ontological modeling with perceptual detection."
Response: We have expanded this discussion in several ways:
Section 3.1 "Philosophical Foundations" now describes COG's approach to visual language compositionality
Section 3.3 includes Table 3 comparing COG with traditional ontological hierarchies, showing the shift from conceptual to perceptual groupings
The distinction between static taxonomies and dynamic, data-driven structures is now emphasized throughout
Comment 3: "Authors need to elaborate on more detailed descriptions of modifications made to YOLOv8."
Response: Section 4.2 "Model Architecture and Training" now provides comprehensive training configuration details including specific hyperparameters, hardware specifications, and the joint detection approach for atomic and contextual COG classes.
Comment 4: "The formal equation effectively captures the detection... would benefit from clearer definitions and explicit representation of relational constraints."
Response: We have enhanced Equation (1) in Section 3.2 with clearer contextual explanation and explicit definitions of components. The equation now includes proper mathematical notation and is better integrated with the surrounding theoretical framework.
Comment 5: "Compare the COG with existing methods designed for understanding diagrams, detecting based on ontologies, or using language models."
Response: We have added Table 2 "Comparison of Visual Understanding Approaches" in Section 2.8 that systematically compares COG with Object Detection (OD), Visual Relationship Detection (VRD), Scene Graph Generation (SGG), and Document Layout Analysis (DLA) approaches across key dimensions.
Comment 6: "Better articulate potential limitations or areas for future work, especially regarding scalability, robustness to visual variations, and real-time implementation challenges."
Response: Section 6.3 "Limitations and Challenges" now provides detailed discussion of orientation/scale invariance issues, OCR dependencies, and two-stage processing requirements. Section 6.4 presents a systematic three-track research roadmap addressing algorithmic enhancement, domain expansion, and system integration.
Comment 7: "Discussing how the current reliance on YOLOv8 influences the generalizability, scalability, and potential limitations."
Response: We have reframed the entire work as a proof-of-concept in Section 6.2 "Proof-of-Concept Achievement vs Future Development Roadmap," which discusses how the current YOLOv8 implementation establishes foundational principles while acknowledging the need for enhanced architectures in future work.
Reviewer 4 Report
Comments and Suggestions for Authors
The manuscript proposes a method called contextual object grouping to facilitate the interpretation of contents in technical security diagrams. The design and the implementation of the proposed method are introduced. The performance is also reported and discussed in the manuscript. The result looks promising.
Some minor comments and suggestions are listed in the following.
1. It is suggested that an example of security digram be provided in Section 1 (Introduction) for readers to better understand the diagram and the challenges to interpret the content based on technologies nowadays.
2. The legend of the symbol might appear in different locations (e.g., the bottom of the diagram). Can the proposed method accurately interpret those diagrams?
Author Response
Comment 1: "It is suggested that an example of security diagram be provided in Section 1 (Introduction) for readers to better understand the diagram and the challenges."
Response: Section 1.1 "Problem Motivation" now includes detailed examples of the challenges in security diagram interpretation, describing how "a red circle may represent a camera in one designer's legend, while another designer uses the same symbol to denote a PIR sensor or electromagnetic lock." This concrete example helps readers understand the fundamental symbol standardization problem that COG addresses.
Comment 2: "The legend of the symbol might appear in different locations (e.g., the bottom of the diagram). Can the proposed method accurately interpret those diagrams?"
Response: This practical consideration is now addressed in Section 6.3 "Limitations and Challenges," where we discuss variable legend configurations and acknowledge this as an area for future enhancement. The modular pipeline architecture described in Section 4.3 is designed to accommodate different legend layouts through spatial relationship analysis.
Reviewer 5 Report
Comments and Suggestions for Authors
This paper addresses the standardization challenge of symbol interpretation in technical security diagrams and proposes the "Contextual Object Grouping (COG)" framework.
- Experiments show that the COG model can only detect symbols in legends, with insufficient recognition ability for isolated or rotated symbols in main diagrams.
- Expand the dataset to 3-5 types of technical diagrams, compare the impact of symbol complexity and layout density on COG performance across different diagram types, and further verify the cross-domain generalization ability of the framework.
- The motivation for writing this paper needs to be further strengthened to improve the content quality of the manuscript. Some existing work is helpful for the manuscript, such as a survey when moving target defense meets game theory and a strategy-making method for PIoT PLC honeypoint defense against attacks based on the time-delay evolutionary game.
- The paper does not mention the real-time indicators of the COG framework, while intelligent sensing scenarios have clear requirements for response speed.
- The discussion section mentions the possibility of combining models such as YOLO-World, but the integration path is not clear.
Author Response
We sincerely thank you for the insightful comments. Below we respond point by point and indicate corresponding revisions in the manuscript.
(1) “COG can only detect symbols in legends; insufficient recognition for isolated/rotated symbols in main diagrams.”
Response. Correct for our proof-of-concept and intentional: we trained COG to recognize symbols strictly in legend context to validate the legend-as-grounding hypothesis. We now state this explicitly and add a short technical roadmap towards robust main-diagram matching (rotation/scale augmentations, rotation-equivariant or oriented heads, and geometric consistency checks). See Discussion → Limitations and Challenges and Future Research Directions.
(2) “Expand to 3–5 diagram types; analyze symbol complexity and layout density; verify cross-domain generalization.”
Response. Agreed. We add a concrete Domain Expansion & Cross-Domain Protocol (P&ID, electrical schematics, network topologies) and define simple strata for symbol complexity and layout density to measure their impact on COG performance. See Discussion → Future Research Directions.
(3) “Strengthen motivation; cite a game-theoretic MTD survey and a PLC honeypoint time-delay evolutionary game paper.”
Response. Added and integrated in the narrative:
Tan et al. (2023), Computer Science Review (game-theoretic MTD survey), and
“Game analysis and decision making optimization of evolutionary dynamic honeypot” (2024), Computers & Electrical Engineering.
They underline that modern defense is adaptive and context-dependent; COG contributes the perceptual substrate by grounding semantics dynamically in the legend. See Introduction and Related Work.
(4) “No real-time indicators; intelligent sensing requires response speed.”
Response. To avoid scope confusion, we removed ‘real-time’ wording and added a neutral footprint description (base model, input size, post-processing type) without claiming streaming performance. The work targets document/diagram analysis; streaming optimization is future work. See Implementation and Methodology.
(5) “Integration path with YOLO-World is unclear.”
Response. We now provide concrete paths:
(i) Two-stage: open-vocabulary detector proposes primitives/text; COG performs legend-as-grounding to attach project-specific semantics.
(ii) Multi-head joint model: shared backbone with an open-vocabulary head and a contextual-grouping head trained with a Legend-Grounding Consistency (contrastive) loss. See Discussion → Future Research Directions.
Reviewer 6 Report
Comments and Suggestions for Authors
This article proposes a novel computer vision framework called Context Object Grouping (COG) for dynamically understanding symbol meanings in technical charts. The core innovation of this framework is to no longer treat legends in charts as auxiliary information that requires post-processing, but to directly learn and recognize symbol label pairs as detectable and unified semantic entities. However, there are still some shortcomings in this article. Here are some suggestions for improvement.
- The methods cited at the beginning of this article should include the latest methods from recent years.
- This article used a custom annotated dataset for training and evaluation, but did not mention whether it was validated on public or cross domain datasets. It is recommended to supplement.
- Although the paper emphasizes the advantages of COG compared to YOLO World, Florence-2 and other models, it does not provide direct comparative experiments with these models. It is suggested to supplement.
- The paper mentions that COG is suitable for real-time intelligent perception applications, but does not provide indicators such as inference speed, model parameter quantity, and computational complexity. It is suggested to increase the discussion.
- This article can further analyze the limitations of other models in dynamic legend learning.
Author Response
We thank the reviewer for the constructive feedback and helpful suggestions. Below we provide point-by-point responses and indicate the concrete changes made in the revised manuscript.
1) “The methods cited … should include the latest methods from recent years.”
Response. We agree. The Related Work section has been clarified and expanded to more explicitly discuss recent advances such as YOLO-World, Florence-2, DetCLIPv3, and DocLLM, and to contrast them directly with COG’s approach. These references were already included in the original submission but have now been more explicitly integrated into the comparative discussion, including a dedicated subsection and a comparative table that highlight how legend-as-grounding differs from vocabulary expansion.
2) “This article used a custom annotated dataset … not validated on public or cross-domain datasets.”
Response. We clarified the scope and added a description of a cross-domain evaluation protocol. This includes additional diagram types such as security layouts, architectural floor plans, P&ID diagrams, electrical schematics, and network topologies. We also define stratification by symbol complexity and layout density to analyze generalization. Furthermore, we committed to releasing a small anonymized test pack (images with legend annotations) upon acceptance, to enable external validation. These additions are described in the Future Research Directions section and in Data Availability.
3) “No direct comparative experiments with YOLO-World, Florence-2, etc.”
Response. While full comparative experiments with large open-vocabulary models are outside the scope of this proof-of-concept, we added a control observation showing that such models correctly name geometric primitives (e.g., “circle”, “arrow”) but cannot derive project-specific semantics from the legend without post-hoc processing. This illustrates the semantic gap that COG addresses. We also defined a comparison protocol for future work, describing common inputs, frozen backbones, shared evaluation tasks, and a Legend-to-Diagram Consistency metric.
4) “The paper mentions that COG is suitable for real-time intelligent perception applications, but does not provide indicators such as inference speed, model parameter quantity, and computational complexity.”
Response. We adjusted the scope of the manuscript. The present work addresses batch document analysis rather than streaming real-time scenarios. We therefore removed references to “real-time” and added a dedicated subsection Footprint description (document setting), specifying the base model, input resolution, and post-processing steps, and explicitly stating that no real-time claims are made. The hardware setup is reported separately in the Training Configuration, and the processing pipeline is described in the Pipeline Architecture. We emphasize that streaming/edge optimization is outside the scope of this work and is planned for future development.
5) “This article can further analyze the limitations of other models in dynamic legend learning.”
Response. We added a dedicated subsection titled Why open-vocabulary models fall short for legend-grounded semantics. This subsection explains three core limitations of current open-vocabulary and multi-task detectors in the context of dynamic legend learning: (i) the absence of local, image-internal grounding, (ii) the lack of compositional binding of symbol+label as a first-class percept, and (iii) the lack of instance-level alignment between legend exemplars and diagram instances. We then describe how COG directly addresses these points by elevating legend symbol–label pairs to detection targets.
Round 2
Reviewer 3 Report
Comments and Suggestions for Authors
In this current version, the authors tried to address most problems that were pointed out during the first round of reviews. The experimental results convincingly demonstrate the effectiveness of the COG framework, with high detection accuracy (mAP50 ≈ 0.99) and perfect symbol-label pairing, validating the core hypothesis that models can learn to perceive semantic groupings as first-class entities directly from data without relying on rule-based post-processing.
However, the comparisons in Table 2 of section 2.8 do not sufficiently support these claims. The table primarily highlights that COG integrates semantic grouping into the detection stage and enables end-to-end detection, contrasting it with existing approaches that rely on post hoc relation inference, external knowledge, or multi-step processing. While these distinctions are informative, they do not explicitly connect the high-performance metrics to the advantages claimed, such as improved contextual understanding, robustness to domain variability, or real-time applicability.
I recommend explicitly connecting the empirical results with the advantages over existing methods. Citing related approaches that have attempted to incorporate semantic or contextual understanding in diagram analysis or object detection could enrich the comparison.
To improve clarity, revise the bullet points and numbering within sub-sections by ensuring consistent indentation, uniform bullet styles or numbering formats, and clear hierarchical structure
Author Response
Dear Reviewer,
We thank you for the constructive feedback and positive assessment of our work. We are pleased that you found our experimental results convincing and that the COG framework's core hypothesis has been validated. Below, we address each of your specific concerns with detailed revisions to strengthen the manuscript.
Note: All modifications from this revision round are marked in blue text to distinguish them from the red revisions made in response to the first review.
Response to Main Concerns
1. Enhanced Empirical Performance Validation
Reviewer Comment: "The comparisons in Table 2 of section 2.8 do not sufficiently support these claims... they do not explicitly connect the high-performance metrics to the advantages claimed."
Our Response: We have addressed your concern by adding a new Table 3 "COG Performance Validation and Empirical Results" that provides dedicated empirical validation metrics alongside the existing theoretical comparison in Table 2. This approach preserves the conceptual framework while offering explicit quantitative support.
The new Table 3 includes:
- mAP50 ≈ 0.99 for comprehensive legend component detection
- 98% symbol-label pairing accuracy (test cases correctly structured)
- Contextual awareness validation demonstrating symbols detected only in legend contexts
- Cross-domain adaptation results across security and architectural diagrams
- Processing efficiency metrics showing single-stage detection without post-processing
This dedicated validation table directly connects our empirical achievements to the theoretical advantages claimed, addressing your concern about the gap between performance metrics and stated benefits.
2. Explicit Connection Between Results and Claimed Advantages
Reviewer Comment: "I recommend explicitly connecting the empirical results with the advantages over existing methods."
Our Response: Following your recommendation, we have added a new section "Quantitative Performance Validation Against Traditional Approaches" at the beginning of Section 6.1. This section provides concrete evidence for COG's advantages through three key areas:
Legend Structure Detection: COG achieves mAP50 ≈ 0.99 compared to traditional approaches that would require separate detection phases plus complex post-processing, with our unified approach achieving 98% symbol-label pairing accuracy.
Contextual Awareness Validation: The model demonstrates true contextual understanding by detecting symbols only within legend contexts (confidence 0.82-0.99), supporting our hypothesis that semantic groupings become first-class perceptual entities.
Dynamic Symbol Interpretation: Successful cross-domain performance without retraining, eliminating manual rule engineering through automatic legend-based semantic mapping.
This quantitative analysis directly supports our theoretical claims about COG's effectiveness in contextually-dependent interpretation scenarios.
3. Enhanced Literature Coverage for Contextual Understanding
Reviewer Comment: "Citing related approaches that have attempted to incorporate semantic or contextual understanding in diagram analysis or object detection could enrich the comparison."
Our Response: In response to your recommendation, we have incorporated a new section "Contextual Understanding in Technical Diagram Analysis" (Section 2.8) that consolidates and contextualizes related approaches attempting semantic or contextual understanding:
Technical Diagram Analysis: We discuss Kalkan et al.'s frameworks for engineering sketch analysis and Lin et al.'s work on drawing restoration, noting how these follow traditional detection-then-interpretation pipelines.
Document Understanding Domain: We examine LayoutLM variants (LayoutLM, LayoutLMv2, LayoutLMv3) and their spatial-semantic integration capabilities, while distinguishing their focus on pre-tokenized relationships from COG's dynamic visual-semantic mapping approach.
Vision-Language Models: We position YOLO-World and Florence-2 as current state-of-the-art contextual detection methods, while explaining how they rely on external vocabulary knowledge rather than learning project-specific symbol semantics from contained legends.
This consolidation better positions COG within the broader landscape of contextual understanding approaches, clarifying how our framework addresses limitations that existing methods cannot overcome.
4. Formatting and Structural Improvements
Reviewer Comment: "To improve clarity, revise the bullet points and numbering within sub-sections by ensuring consistent indentation, uniform bullet styles or numbering formats, and clear hierarchical structure."
Our Response: We have reviewed the manuscript and improved formatting consistency throughout, including standardized table formatting, consistent use of \itemize for most lists, and retention of \enumerate only where sequential numbering is essential for content comprehension (such as the Contributions section).
Note on Formatting Changes: These formatting improvements have been implemented without color coding to avoid obscuring the substantive content modifications, which remain clearly marked in blue for your review.
These improvements enhance readability and maintain the professional presentation standards expected for academic publications while preserving content-appropriate formatting choices.
Summary of Revisions
We have systematically addressed your concerns through the following modifications (marked in blue to distinguish from previous red revisions):
- Added Table 3 with dedicated empirical validation metrics that complement theoretical comparisons in Table 2 (Section 2.9)
- Incorporated quantitative performance validation section that explicitly connects our results to claimed advantages (Section 6.1.1)
- Consolidated contextual understanding literature into a dedicated section that positions COG within existing approaches (new Section 2.8)
- Standardized formatting consistency across tables, lists, and structural elements while maintaining content-appropriate choices (Sections 1.4, 4.3, and table formatting throughout)
These revisions respond comprehensively to your feedback while preserving the manuscript's focus on demonstrating COG as a foundational framework for context-aware sensing applications. The blue-colored additions clearly distinguish this revision round from our previous red modifications, facilitating your review of our responses.
We believe these modifications fully address your concerns and strengthen both the empirical validation and contextual positioning of our work within the intelligent sensing and contextual object detection literature.
Thank you again for your constructive feedback, which has substantially improved the clarity and impact of our contribution.
Sincerely, The Authors
Reviewer 5 Report
Comments and Suggestions for Authors
accepted
Round 3
Reviewer 3 Report
Comments and Suggestions for Authors
The authors have addressed most of my previous review comments satisfactorily and the paper has improved significantly. Some figures present a poor resolution. Consider using vectorial format instead of raster format to improve their quality.
Author Response
We sincerely thank you for the insightful comment and suggestions: Some figures present a poor resolution. Consider using vectorial format instead of raster format to improve their quality.
Response: Since our experiments are inherently based on raster images, the options for full vectorization are limited. However, in the revised version we improved figure quality by regenerating all layout-related images at higher resolution (300 DPI instead of original 72 DPI), ensuring substantially clearer presentation in the final manuscript.